# The Prevalence and Risk Factors of Type 2 Diabetes Mellitus (DMT2) in a Semi-Urban Saudi Population

**DOI:** 10.3390/ijerph17010007

**Published:** 2019-12-18

**Authors:** Mohammed Abdullah Al Mansour

**Affiliations:** Department of Family Medicine, College of Medicine, Majmaah University, Almajmaah 11952, Saudi Arabia; m.aalmansour@mu.edu.sa

**Keywords:** diabetes mellitus, prevalence, risk factors, semi-urban population

## Abstract

(1) Background: Diabetes mellitus is a common health problem in Saudi Arabia, causing a huge burden for individuals, families, and communities. The objectives of the current study were to determine the prevalence and risk factors of type 2 diabetes mellitus among a semi-urban population of Saudi Arabia. (2) Research methods: The research design was cross-sectional, and the research was conducted in five primary health care centers (PHCC) in Majmaah, Saudi Arabia. The sample size was calculated as 353. A pre-tested questionnaire was used to collect data after obtaining ethical approval. Blood samples were taken to assess glucose levels and other variables. SPSS version 21 was used to analyze data. (3) Results: The prevalence of type 2 diabetes mellitus was 34.6%. The disease was more prevalent among the older respondents compared with the younger age groups (44.6% versus 15.6%). We found that females acquire the disease at a slightly higher rate than males (34.9% versus 34.2%), but this difference is not statistically significant. The sociodemographic risk factors of the disease were as follows: old age (44%), business and private occupation (38.5%), divorced or widowed (56.3%), and low income (42.4%). The health behaviors factors were as follows: overweight or obese status (42.3%), high triglycerides (TG) (43.4%), low high-density lipoprotein (HDL) (37.3%), and high total cholesterol (23.7%). There was a statistically significant difference in these risk factors between patients with and without diabetes. (4) Conclusion: The prevalence of type 2 diabetes mellitus among the semi-urban population of Saudi Arabia is high. The disease is more prevalent among elderly respondents and is associated with obesity, high TG, low HDL, and high total cholesterol.

## 1. Introduction

Diabetes mellitus is a multifactorial disease of considerable heterogeneity [1,2,3]. The disease is the most common chronic endocrine disorder, affecting an estimated 5%–10% of adults worldwide [4,5]. Predictions based on many studies have indicated a growing increase of diabetes mellitus, particularly in developing countries. It is predicted that between 2010 and 2030, developed and developing countries will see a 20% and 69% increase, respectively, in the number of adults with diabetes [6]. The prevalence of diabetes among those aged 20–79 years may increase to 7.7%, constituting 439 million by 2030 [7]. Many studies have shown that lifestyle modifications are effective in preventing obesity and diabetes in high-risk adults with impaired glucose tolerance [8].

Globally, the east Mediterranean region has the second highest prevalence of diabetes in terms of population. According to the World Health Organization (WHO), almost a quarter of the region’s population has diabetes of one form or another [2].

Data from the Gulf revealed high prevalence rates of type 2 diabetes mellitus. The reported prevalence rates were 25.7%, 16.1%, and 21% in Bahrain, Oman, and Kuwait, respectively [8,9,10].

The prevalence of diabetes mellitus is high in the Saudi population and represents a major public health concern; the number of diabetic patients has steadily increased over the last several decades [11]. The prevalence of diabetes mellitus in Saudi Arabia was found to be 31.6% of the general population, 14.1% of the working population, 34.6% of males, and 27.6% of females [4,12,13]. Diabetes mellitus has a high economic impact; the indirect national financial burden is likely to exceed 0.87 billion USD, and patients with diabetes, on average, incur healthcare costs that are almost ten times higher ($3686 vs. $380) than those without diabetes [14,15].

The objective of the current study was to determine the prevalence and risk factors of type 2 diabetes mellitus in a semi-urban Saudi population.

## 2. Patients and Methods

The cross-sectional study was conducted in Majmaah City, Riyadh Province, Saudi Arabia [16]. Patients of both sexes who attended primary health care centers (PHCCs) during the study period were enrolled in the study. Children, non-Saudi patients, and those who were nonresidents of Majmaah were excluded from the study. At the level of the PHCC, stratified random sampling was employed; five out of the 11 PHCCs in the city were selected. Stratification was based on the patients’ load. The selection of the respondents at the level of PHCCs was performed by systematic sampling, and the interval was counted by dividing the estimated average number of patients attending the center every day by the number of the sample agreed to be taken in the same day.

A total of 384 patients were enrolled in the study, of which 353 completed the study. The sample size was calculated using the level of precision formula (*n* = Z2 × *p* (1 − *p*)/d2), where *n* is the sample size, z is the standard error associated with the chosen level of confidence (1.96), *p* is estimated prevalence (0.50), q is 1 − *p* (0.50), and d is the acceptable error (0.05). The patients were interviewed with a structured questionnaire. Patients’ demographic data, medical history, and social and lifestyle history were documented. Blood pressure was taken in a single visit by an electronic sphygmomanometer. Joint National Committee 8 (JNC 8) criteria were taken as standard in the measurements [17]. High blood pressure readings were defined as ≥ 140/90 mm of Hg. Height and weight were measured using calibrated portable scales and a steel tape measure, respectively. Patients were weighed with light clothes and with shoes off. The height of patients was measured without shoes. The formula weight (kg)/height (m)2 was used to calculate the BMI of the respondents. Respondents who scored a BMI of 25–29.9 were considered overweight, and those who had a BMI of 30 or more were considered obese [18].

Blood samples were extracted immediately if the patient was fasting or on the following day if not in order to test fasting blood glucose and HBA1c. Respondents who met the ADA 2014 criteria of diabetes mellitus were diagnosed as patients with diabetes [19]. Blood samples to test triglycerides (TG), total cholesterol, high-density lipoprotein (HDL), and low-density lipoprotein (LDL) were also taken from all respondents. The cut-off levels were used according to the suggestions of the National Cholesterol Education Program Adult Treatment Panel III [20].

The data were entered and analyzed using SPSS for Windows, version 21 (SPSS, Chicago, IL, USA). Descriptive statistics were used, and comparisons between qualitative data were made using chi-square tests to gauge significance. A p value of less than 0.05 was considered statistically significant.

Ethical approval was obtained from the Majmaah University Ethics Committee. All the respondents gave their written informed consent, all data were kept confidential, and the right of the patients to withdraw from the study was respected throughout.

## 3. Results

The results show that 122 (34.6%) of the respondents had diabetes mellitus, as shown in Figure 1.

Table 1 shows the relationship between diabetes mellitus and various sociodemographic characteristics. Nineteen (15.6%) of the respondents younger than 40 years of age and 103 (44.6%) of those at least 40 years old had diabetes mellitus (*p* < 0.001). A total of 56 (34.2%) and 66 (34.9%) of males and females had the disease, respectively. Diabetes mellitus rates among business or private persons, government employees, housewives, and students were 38.5%, 32%, 10.3%, and 31.9%, respectively. Five (10.4%) of the single, 99 (36.3%) of the married, and 18 (56.3%) of the divorced or widowed respondents had type 2 diabetes mellitus (*p* < 0.001). The disease in the low-, moderate-, and high-income groups constituted 42.4%, 29.0%, and 26.1%.

Table 2 shows the risk factors of diabetes mellitus. Performing regular physical activities, smoking tobacco, consuming fatty foods, high LDL, and high blood pressure were statistically insignificant factors in patients with diabetes. On the other hand, seven (13.2%) of the normal or underweight respondents had type 2 diabetes mellitus compared to 115 (42.3%) of the overweight or obese patients (*p* < 0.001). Triglyceride levels were found to be influential, as 76 (32.5%) patients with desirable or borderline TG had type 2 diabetes mellitus, compared with 46 (43.4%) patients with high levels of triglycerides (*p* < 0.004).

Total cholesterol levels were also found to be significant, as the results showed that 113 (35.9%) patients with desirable total cholesterol levels and nine (23.7%) patients with high levels had type 2 diabetes mellitus (*p* < 0.016). A total of 101 (37.3%) patients with low/average HDL and 21 (25.6%) patients with HDL had DMT2 (*p* < 0.012).

The logistic regression results are presented in Table 3. The risk of getting diabetes lowered as respondents’ ages decreased (adjusted odds ratio (AOR) = 0.536, *p* < 0.001). The rest of the factors were not significant (*p* > 0.05). Patients who had DMT2 had significantly high levels of triglycerides (AOR = 0.654, *p* = 0.004) and high BMI (AOR = 0.667, *p* = 0.013). These patients with diabetes had significantly low levels of HDL (AOR = 1.540, *p* = 0.012) and total cholesterol (AOR = 2.109, *p* = 0.016). 

## 4. Discussion

The data were collected from 353 patients who attended a PHCC in Majmaah, Saudi Arabia. According to our findings, the prevalence of DMT2 was 34.6%. This rate is higher than results from Egypt (11.4%), Yemen (3.0%), Iraq (10.2%), and Algeria (8.5%) [21]. The prevalence of DMT2 is also higher than the findings elsewhere in Saudi Arabia [5,12,13]. As Majmaah is a semi-urban area, this finding is unexpected; however, it may be explained by the fact that the city has experienced urbanization in the last few years. The establishment of Majmaah University and Sudair Industrial City has brought manpower from abroad and created new jobs and financial investment opportunities, resulting in lifestyle changes and the improved economic status of those in the population. having this in mind, the higher rate of DMT2 may be accounted for by the sample, which was taken from patients in PHCCs and not from the general population.

Our finding shows a significant relationship between DMT2 and age. Patients aged forty or older are more likely to have the disease compared with the younger age group (*p* < 0.001), which is congruent to the results of Alnozha et al. [5].

In the current study, females showed a slightly higher prevalence rate of the disease than males (34.9% vs. 34.2%, *p* = 0.9164). This finding is consistent with studies conducted in Saudi Arabia and Iran [5,22] and is inconsistent with studies done in Saudi Arabia and France, where DMT2 was more prevalent in males [12,13,23].

DMT2 shows significant association with occupation (*p* < 0.001) [24,25]. Business or private personnel showed higher prevalence of DMT2 (38.5%) compared with government employees (32%), students (31.9%), and housewives (31.9%).

In this study, we found a higher prevalence of DMT2 among married and divorced or widowed respondents compared with single respondents (*p* < 0.001). These findings are consistent with those of Murad et al. [23]. Marriage affects lifestyle; couples may increase food intake and become less active after marriage, leading to increased body weight and risk of developing the disease [23,25]. The prevalence of DMT2 does not show significant association with economic status [26,27]. Although the results of this study show a nonsignificant association between DMT2 and performing regular physical activities (35.7% vs. 31.1%, *p* = 0.336), the physical activities were shown in other studies to play a role in the development of diabetes [23,28,29,30,31].

Respondents who consumed fatty foods on a daily basis had a higher prevalence of DMT2 compared with those who did not [32]. Still, the relationship between DMT2 and consuming fatty foods on a daily basis is not significant.

Our study shows that tobacco smokers had DMT2 more than nonsmokers [2,33], but the relationship is not significant. This finding is inconsistent with that of Thelin et al. [2]. High levels of triglycerides, total cholesterol, and HDL showed significant associations with DMT2, in agreement with previous studies [2,34]. The relationships of these factors with DMT2 are consistent allover lipid profile studies. The variability of lipid panel components are noticed in this study as in the study of Alkaabba et al., where the percentages of total cholesterol were almost the same. The significance of total cholesterol as a risk factor is, however, different in this study [35], which shares the same results as an Iranian study [22].

The weight status in the current study shows significant association with DMT2, congruent to other studies where the prevalence of DMT2 was higher among obese subjects [36,37,38]. The prevalence of DMT2 tends to be higher among hypertensive patients, but in this study, we found that hypertension’s relationship to DMT2 is insignificant. This finding is inconsistent with that of El-Hazmi et al. [39]. This might be explained by our method of taking blood pressure measurements; taking one reading in one setting is usually not conclusive of blood pressure status [17].

The study draws attention to the fact that DMT2 is no longer a disease of urban communities. More social studies need to be conducted to explore the root causes of this shift at the community level. Future researches should be done, in prospective longitudinal studies, in order to explore risk factors and their association or causation effects on the prevalence of DMT2 in smaller cities and how this reflects on the wider community.

## 5. Conclusions

The prevalence of type 2 diabetes mellitus among the semi-urban population of Saudi Arabia tested in this study is high. The disease is more prevalent among elderly respondents and is associated with obesity, high level of triglycerides, low HDL, and high total cholesterol. A comprehensive approach is needed to promote healthy lifestyles and avoid the burden of urbanization’s negative health behaviors in communities of this size. 

## Figures and Tables

**Figure 1 ijerph-17-00007-f001:**
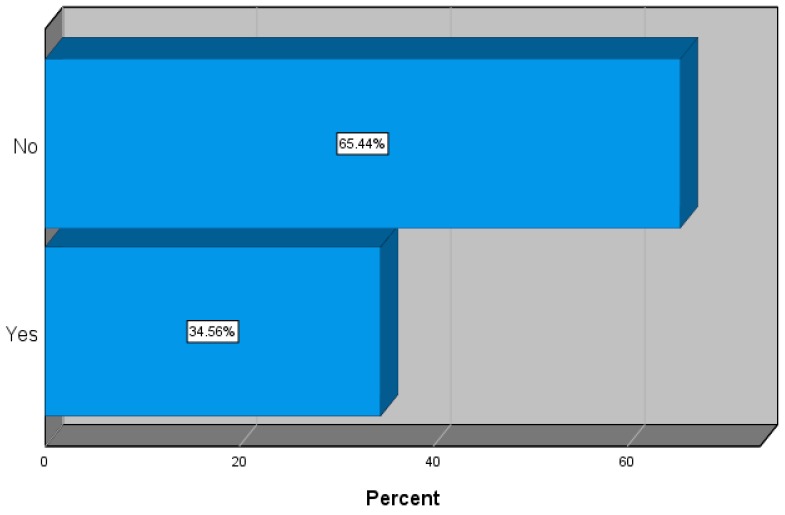
Prevalence of type 2 diabetes mellitus (DMT2).

**Table 1 ijerph-17-00007-t001:** The relationship between DMT2 and sociodemographic characteristics (*n* = 122).

Sociodemographic Factors	Diabetes Mellitus		Total	*p*
	Present	Absent		
Age/years:				<0.001
Less than 40	19 (15.6%)	103 (84.4%)	122 (34.6%)	
40 and more	103 (44.6%)	128 (55.4%)	231 (65.4%)	
Gender:				0.9164
Male	56 (34.2%)	108 (65.8%)	164 (46.5%)	
Female	66 (34.9%)	123 (65.1%)	189 (53.5%)	
Occupation:				<0.001
Business or private	57 (38.5%)	91 (61.5%)	148 (41.9%)	
Government employees	41 (32%)	87 (68%)	128 (36.3%)	
Housewife	3 (10.3%)	26 (89.7%)	89 (8.2%)	
Students	7 (31.9%)	15 (68.1%)	22 (6.2%)	
Others	14 (53.8%)	12 (46.2%)	26 (7.4%)	
Marital status:				˂0.001
Single	5 (10.4%)	43 (89.6%)	48 (13.6%)	
Married	99 (36.3%)	174 (63.7)	273 (77.3%)	
Divorced/widow	18 (56.3%)	14 (43.7%)	32 (9.0%)	
Monthly income (SR):				0.032
Low (<5000)	70 (42.4%)	95 (57.6%)	165 (46.7%)	
Moderate (5000–10,000)	29 (29%)	71 (71%)	100 (28.3%)	
High (>10,000)	23 (26.1%)	65 (73.9%)	88 (25.0%)	

**Table 2 ijerph-17-00007-t002:** Risk factors of DMT2.

Risk Factor	Diabetes Mellitus		Total	*p*
	Present	Absent		
Perform regular physical activity:				0.490
Yes	28 (31.1%)	62 (68.9%)	90 (25.5%)	
No	94 (35.7%)	169 (64.3%)	263 (74.5%)	
Total	122 (34.6%)	231 (65.4%)	353 (100%)	
Consumption of fatty foods:				0.248
Yes	108 (35.6%)	195 (64.4%)	303 (85.8%)	
No	14 (28.0%)	36 (72.0%)	50 (14.2%	
Total	122 (34.6%)	231 (65.4%)	353 (100%)	
Tobacco smoking:				0.132
Yes	9 (25.7%)	26 (74.3%)	35 (9.9%)	
No	113 (35.5%)	205 (64.5%)	318 (90.1%)	
Total	122 (34.6%)	231 (56.4%)	353 (100%)	
BMI:				0.013
Underweight/normal	7 (13.2%)	46 (86.8%)	53 (15.0%)	
Overweight/obese	115 (42.3%)	185 (57.7%)	300 (22.1%)	
Total	122 (34.6%)	231 (65.4%)	353 (100%)	
Low-density lipoprotein (LDL):				0.869
Optimum/Border line	95 (34.8%)	178 (65.2%)	109 (85.4%)	
High	27 (33.7%)	53 (66.3%)	80 (22.6%)	
Total	122 (34.6%)	231 (65.4%)	353 (100%)	
Triglycerides:				0.004
Desirable/borderline	76 (32.5%)	158 (67.5%)	234 (66.3%)	
High	46 (43.4%)	73 (56.6%)	119 (33.7%)	
Total	122 (34.6%)	231 (65.4%)	353 (100%)	
Total cholesterol:				0.016
Desirable	113 (35.9%)	202 (64.1%)	315 (89.2%)	
High	9 (23.7%)	29 (76.3%)	38 (10.8%)	
Total	122 (34.6%)	231 (65.4%)	353 (100%)	
High-density lipoprotein (HDL):				0.012
Low/average	101 (37.3%)	170 (62.7%)	271 (76.8%)	
High	21 (25.6%)	61 (74.4%)	82 (23.2%)	
Total	122 (34.6%)	231 (65.4%)	353 (100%)	
Hypertension:				0.221
Normal	111 (33.9%)	216 (66.1%)	327 (92.6%)	
High	11 (42.3%)	15 (57.7%)	26 (7.4%)	
Total	122 (34.6%)	231 (65.4%)	353 (100%)	

**Table 3 ijerph-17-00007-t003:** Logistic regression.

Sociodemographic Characteristics				
**Item**	**AOR**	***p***	**95% CI for AOR**	
			**Lower**	**Upper**
Age	0.536	<0.001 *	0.425	0.675
Gender	1.152	0.609	0.669	1.983
Occupation	0.208	0.999	0.880	1.137
Monthly income	0.890	0.214	0.897	1.629
**Risk Factors**				
**Item**	**AOR**	***p***	**95% CI for AOR**	
			**Lower**	**Upper**
Regular physical exercise	0.768	0.338	0.448	0.675
Fatty food	1.205	0.596	0.604	2.402
smoking	0.516	0.122	0.223	1.195
BMI	0.667	0.013 *	0.485	0.918
LDL	0.906	0.329	0.742	1.105
Total cholesterol	2.109	0.016 *	1.152	3.861
HDL	1.540	0.012 *	1.098	2.161
Triglyceride	0.654	0.004 *	0.491	0.871

* Significant at 5% level of significance; AOR: Adjusted odds ratio.

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
