# Peer review of "The Prevalence and Risk Factors of Type 2 Diabetes Mellitus (DMT2) in a Semi-Urban Saudi Population"

_ijerph, 2019, doi:10.3390/ijerph17010007_

Round 1
Reviewer 1 Report
Reviewed manuscript entitled "Prevalence and risk factors of Type 2 Diabetes Mellitus (DMT2) in a semi-urban Saudi population" contains results that are similar to results obtained by other authors (for example from Europe). Therefore manuscript needs additional parts. There are also necessary changes.
Minor revision:
1. Line 17. ".... the disease more than the males..." Are statistical important differences between values for females (34,9%) and for males (34,2%)?
2. Line 57. Explain please formula for calculation of sample size. What is this: Z, p, g, d? This calcualtion needs more details.
3. Value of BMI is less important. It should be included waist-hip ratio (WHR). This value shows type of obesity.
4. Line 84. It is "socio demographic" > "socio-demographic"
5. Line 87. It is "sixty - six" > "sixty-six"
6. References must be arranged. Compare for example list of authors in Ref. 19 and 23 or 19 and 22. There are also different forms used for cited publications (compare for example 17 and 27). References should be according Instruction for Authors.
Major revision
Discussion is too poor. Autor should compare obtained results to similar results from other countries. In Discussion must be included suggestions, why, for example, are differences in prevalence of DM in dependence on age, gender and other investigated socio-demographic factors. Discussion must be more rich. It should be also included value of fasting glucose, very important factor for DM.
Author Response
Dear reviewer
am glad with your valuable comments
please check the modified version of the manuscript
i tried to incorporate all of your comments
appreciating your support
regards

Reviewer 2 Report
Thank you for the opportunity to review this manuscript. The following comments and questions are submitted with all due respect to the authors.
Abstract
Line 12: Consider rewording sentence that refers to the sample size.
Line 12-13: By "pre-tested questionnaire," are the authors referring to a validated questionnaire?
Line 14: Was SPSS version 21 "used" to analyze the data? Consider re-wording.
Line 16: The prevalence rates for males and females are quite similar (34.9 vs. 34.2). This does not appear to justify the statement, "females acquire the disease more..."
Line 21: Consider rewording "old age group"
Line 21-22: The factors mentioned here are different than the factors listed in Lines 17-19. It may be helpful to review and clarify when sociodemographic factors OR health behaviors/factors that can place one at higher risk are being referred to.
Introduction
Line 29 and 31: Rather than make definitive statements ("there will be" and "will increase"), consider "it is predicted that" or "may increase"
Line 40, 44, 45: Consider stating "patients with diabetes" instead of diabetic patients and "patients without diabetes" instead of non-diabetics.
Materials and Methods
Line 50-51: The word "who" may be missing between "patients" and "attended"
Line 68: Consider stating "patients with diabetes" or "people with diabetes" instead of diabetics
-Please review grammar.
Results
Figure 1: This information may be better presented in text, rather than dedicating a figure to it.
Table 1: Review format for consistency
Lines 84-91: Consider which information from Table 1 you would like to highlight in text vs. which information is best to present in the table only.
Lines 93-106: Consider which information from Table 2 you would like to highlight in text vs. which information is best to present in the table only.
Line 112: Consider stating "patients with diabetes" or "people with diabetes" instead of diabetics
Discussion
Lines 125-127, 132-134: Do the authors have some insight associated with this finding?
Line 150: Consider "weight status" instead of "nutritional status."
-Please review grammar.
Conclusions
-The authors may wish to address both health behaviors/factors and sociodemographic factors.
-Please review grammar.
Author Response

(The authors gave the same response as above.)

Reviewer 3 Report
The objective of the current study was to determine the prevalence and risk factors of Type 2 Diabetes Mellitus in a semi-urban Saudi population.
The sample size was taken as 353 calculated by the formula (n=Z2 ×p q/d2). The authors should indicate the values they have used in the formula.
The inclusion and exclusion criteria are not clear. You must include them in the material and methods section.
It seems that the study was conducted on all patients who went to these 5 health centers in one morning and agreed to participate. This produces a very important bias. It does not seem to be a representative sample of the general population of the area, nor does it follow the population that usually goes to the health center.
Was the study also performed on the patient who was already diagnosed with diabetes?
Were patients with active corticotherapy or severe active inflammatory process excluded?
A sample of blood to test fasting blood glucose was taken from participants to diagnose diabetes. To be more sensitive the diabetes diagnostic tests, you should have also used the HbA1c determination or an oral glucose overload test.
Fundamental anthropometric parameters such as abdominal circumference should have been collected.
Figure 1 does not contribute anything.
Table 1 shows the concept of "Monthly income" that is not explained at the bottom of the table or in materials and methods.
It is not clear how patients are identified by "Consumption of fatty foods". Do they ask the patient directly? In that case it is a subjective assessment of little value. Do you conduct a dietary survey?
The conclusion of the study is that Diabetes is more prevalent among old age group, is associated with obesity, High Density lipoprotein, triglycerides and Total Cholesterol. It does not contribute anything that is not known
Author Response

(The authors gave the same response as above.)

Round 2
Reviewer 3 Report
I believe that the article is acceptable for publication after the last revision